# Impact of HLA-DR Antigen Binding Cleft Rigidity on T Cell Recognition

**DOI:** 10.3390/ijms21197081

**Published:** 2020-09-25

**Authors:** Christopher Szeto, Joseph I. Bloom, Hannah Sloane, Christian A. Lobos, James Fodor, Dhilshan Jayasinghe, Demetra S. M. Chatzileontiadou, Emma J. Grant, Ashley M. Buckle, Stephanie Gras

**Affiliations:** 1Department of Biochemistry and Molecular Biology, Biomedicine Discovery Institute, Monash University, Clayton, VIC 3800, Australia; chris.szeto@monash.edu (C.S.); joseph.i.bloom@gmail.com (J.I.B.); hslo1@student.monash.edu (H.S.); christian.lobos@monash.edu (C.A.L.); fods12@gmail.com (J.F.); dhilshan.jayasinghe@monash.edu (D.J.); dimitra.chatzileontiadou@monash.edu (D.S.M.C.); emma.grant@monash.edu (E.J.G.); ashley@ptngconsulting.com (A.M.B.); 2Eccles Institute of Neuroscience, John Curtin School of Medical Research, The Australian National University, Canberra, ACT 0200, Australia; 3Australian Research Council Centre of Excellence for Advanced Molecular Imaging, Monash University, Clayton, VIC 3800, Australia

**Keywords:** human leukocyte antigen (HLA), MHC class II, peptide binding, protein stability, molecular dynamics, ensemble refinement, TCR binding

## Abstract

The interaction between T cell receptor (TCR) and peptide (p)-Human Leukocyte Antigen (HLA) complexes is the critical first step in determining T cell responses. X-ray crystallographic studies of pHLA in TCR-bound and free states provide a structural perspective that can help understand T cell activation. These structures represent a static “snapshot”, yet the nature of pHLAs and their interactions with TCRs are highly dynamic. This has been demonstrated for HLA class I molecules with in silico techniques showing that some interactions, thought to stabilise pHLA-I, are only transient and prone to high flexibility. Here, we investigated the dynamics of HLA class II molecules by focusing on three allomorphs (HLA-DR1, -DR11 and -DR15) that are able to present the same epitope and activate CD4+ T cells. A single TCR (F24) has been shown to recognise all three HLA-DR molecules, albeit with different affinities. Using molecular dynamics and crystallographic ensemble refinement, we investigate the molecular basis of these different affinities and uncover hidden roles for HLA polymorphic residues. These polymorphisms were responsible for the widening of the antigen binding cleft and disruption of pHLA-TCR interactions, underpinning the hierarchy of F24 TCR binding affinity, and ultimately T cell activation. We expanded this approach to all available pHLA-DR structures and discovered that all HLA-DR molecules were inherently rigid. Together with in vitro protein stability and peptide affinity measurements, our results suggest that HLA-DR1 possesses inherently high protein stability, and low HLA-DM susceptibility.

## 1. Introduction

T cell-mediated immunity is dependent on the activation of CD4+ and CD8+ T cells, which are responsible for cytokine secretion and induction of apoptosis within infected cells [1]. T cells are activated upon T cell receptor (TCR) recognition and binding of major histocompatibility complexes (MHC) presenting antigenic peptides (p) on the surface of infected cells [2]. The classical MHC molecules in humans, called HLA (Human Leukocyte Antigens), are divided into class I (HLA-A, -B, -C) and class II (HLA-DR, -DQ, -DP) molecules, which are recognized by CD8+ and CD4+ T cells, respectively [3]. Due to high polymorphism, HLAs can present a vast range of peptides from both foreign and self-proteins recognised by T cells. This ensures that the immune system efficiently recognises a large range of infectious pathogens.

Peptides presented within the peptide binding cleft are sandwiched between two α-helices that form the walls of the cleft and sit on top of a β-sheet floor. Although the protein subunits that make up HLA class I (HLA-I, heavy α-chain and β2-microglobulin) and HLA class II molecules (HLA-II, α- and β-chains) are different, they share similar secondary structures. Their main differences occur at the ends of the binding clefts, where both ends of the cleft are closed-off in HLA-I but open-ended in HLA-II. The spatial dimensions within the binding cleft of HLAs define the way in which peptides are anchored and presented. The structural arrangement of the closed-off HLA-I favours the binding of smaller peptides, typically 8–10 amino acids in length, whilst the open-ended HLA-II often binds longer peptides (>11 residues). Peptides presented by HLA-II molecules sit neatly across the binding cleft in four distinct pockets (P1, P4, P6 and P9) [4], in which the anchor residues bind, and overhanging peptide residues protrude outwards from the binding cleft. Other differences include the α1-helix of HLA-I molecules being continuous and spanning the length of the presented peptide [5], whereas, the α-helix of the HLA-II α-chain (residues 45–79) is discontinuous and possesses an N-terminal kink (residues 52–55), forming a short intermittent unstructured loop, which can stabilize protruding N-terminal residues (P-2 and P-1) [4].

HLA-II are unstable during peptide loading in endosomes (pH 4.5–6), where the molecular chaperone HLA-DM directly facilitates the dissociation of class II-associated invariant chain peptide (CLIP) or low affinity peptides by inducing conformational changes and stabilising HLA-II via residues from the P1 pocket [6]. The loading of a peptide requires strong interactions between peptide and HLA-II, as the peptide needs to displace HLA-DM. During the translocation of pHLA-II from endosome to cell surface, the complex transitions from a malleable and “floppy” state, into a highly stable and “compact” structure as it progresses from an acidic to a neutral pH environment [7,8,9]. From the limited reports of thermal stability profiles, the melting temperatures (T_m_) for pHLA-II molecules at neutral pH exceed those observed for pHLA-I molecules [10,11]. Mutagenesis studies that introduced mutations that stabilise the P1 anchor residue or the P1 pocket led to increases in thermal stability [11]. Mutations in P1, P4, P6 and P9 also influence HLA-II peptide dissociation rates, with longer half-lives correlating with immunodominance [12]. Collectively, these studies suggest that pHLA-II stability is correlated with high affinity peptide binding, and in turn an effective T cell response.

A recent investigation using molecular dynamics (MD) and crystallographic ensemble refinement techniques discovered that a large proportion of pMHC class I complexes exhibited conformational variation in both peptide and MHC residues stabilising the pMHC complex [13]. The study shows that pMHC-I are dynamic, and that a single snapshot of the TCR-pMHC-I interaction may not be sufficient to fully understand the intricacies of TCR recognition and T cell activation. As pMHC-II are associated with high stability at physiological pH levels, it was of interest to investigate their intrinsic flexibility and its impact on T cell recognition. We recently solved the crystal structures of three HLA-DR molecules (HLA-DRB1 *01:01, HLA-DRB1 *11:01, HLA-DRB1 *15:02, abbreviated to HLA-DR1, -DR11, -DR15, respectively) presenting the 13-mer core epitope of Gag_293_ peptide derived from the HIV gag protein (Gag_299–311_, RQ13), free and bound to the same F24 TCR [14]. This F24 TCR is able to recognise all three HLA-DR-RQ13 complexes despite polymorphisms in the HLA-DRβ chain, which impacts the size of the binding cleft, with HLA-DR1 and HLA-DR15 having a more open cleft than HLA-DR11. F24 TCR-HLA-DR-RQ13 structures revealed the same mode of TCR recognition with the closing of the binding cleft for HLA-DR1 and HLA-DR15, which correlated with lower affinity and different binding kinetics relative to HLA-DR11. Using these crystal structures, we performed MD and ensemble refinement to understand whether HLA-DR polymorphism could impact pHLA-II flexibility, and how their structural differences impact F24 TCR recognition. Our MD results showed structural rigidity within all three HLA-DR clefts, whilst ensemble refinement models revealed polymorphic residues that were responsible for the opening of the HLA-DR cleft and its closing upon F24 TCR binding. As HLA-II molecules are malleable during peptide loading but stable at physiological pH levels, taken together, our results suggest that HLA-DRβ polymorphisms do not influence the flexibility of the cleft in this case, but rather, that polymorphisms can reshape the HLA-DRβ cleft impacting on T cell recognition.

To ensure that our observations were not specific to the RQ13 peptide, we expanded our ensemble refinement analysis to measure the structural variability of all available pHLA-DR structures. We found that for the majority of HLA-DR molecules, the cleft is rigid. We further determined that the open cleft conformation of HLA-DR1 did not affect the flexibility of the pHLA complex, but was associated with higher stability and lower HLA-DM susceptibility compared to HLA-DR11. These findings might have implications for the nature of the peptide repertoire presented by HLA-DR1, compared to other HLA-II, as well as for T cell recognition of HLA-II molecules.

## 2. Results

We used MD and crystallographic ensemble refinement to study the dynamics of HLA-DR1, -DR11 and -DR15. Whilst MD involves purely computational simulation starting from experimentally-determined structures, ensemble refinement extends the usual practice of refining against a set of experimentally-derived crystallographic reflections to produce a single protein structure by combining the reflection data with short, steered MD simulations [15,16,17]. This produces an ensemble of structurally-distinct models of the protein, often improving the refinement statistics (i.e., free R-factor). Ensemble refinement then becomes useful for understanding the dynamics that are consistent with the experimentally determined X-ray data and enables us to explore the conformational landscape (accessible protein conformations) near the solved structure. By contrast, MD is not restricted by the X-ray data but based on approximations of Newtonian physics to derive protein dynamics. MD simulations can therefore be used to observe larger scale dynamics that are not possible due to the cryo-cooled temperatures of crystallographic data collection.

Consequently, both ensemble refinement and MD were used to understand the basis of the structural differences between HLA-DR-RQ13 complexes and F24 TCR affinity. How might dynamics at the level of each individual amino acid derived from ensemble refinement impact the HLA-DR cleft in HLA-DR11, HLA-DR15 and HLA-DR1? Consequently, how might dynamics derived from MD influence our understanding of cleft structural flexibility or rigidity upon binding of the RQ13 peptide? A highly dynamic conformational space (high flexibility) for the binding cleft might reduce the significance of previously observed structural differences, whereas high rigidity infers that the HLA polymorphism impacts on T cell recognition and drives the difference observed in TCR affinities.

### 2.1. Peptide Binding Cleft Rigidity of HLA-DR-RQ13 Complexes Revealed by Molecular Dynamics

We previously solved the structures of the Gag protein-derived RQ13 peptide in complex with the HLA-DR11, -DR15, and -DR1 molecules [14]. The superimposition of the three HLA-DR-RQ13 structures revealed an opening of the peptide binding cleft at the apex of the HLA-DRβ α-helix in HLA-DR15 and HLA-DR1 relative to HLA-DR11 (Figure 1A). The HLA-DRα chain is invariant and as the HLA-DR-RQ13 complexes presented the same peptide, these structural differences were attributed to the polymorphisms within the HLA-DRβ chain. Between these three allotypes, there are 20 polymorphic residues, in which we show that 11 directly impact peptide presentation and are arrayed on the C-terminal β-sheet floor and along the apex of the HLA-DRβ α-helix (Appendix A). We have also determined the structures of a single TCR, F24 TCR, in complex with the three HLA-DR-RQ13 complexes [14]. Comparison of the F24 TCR-HLA-DR-RQ13 complex structures revealed that all three HLA-DR molecules have a closed binding cleft (Figure 1B). The magnitude of the shift for HLA-DR15 and HLA-DR1 when bound to F24 TCR correlated with a lower affinity, slower on-rate and faster off-rate relative to F24 TCR binding to HLA-DR11 [14]. To investigate whether the closing of the cleft was due to the intrinsic flexibility of the HLA-DR molecule (conformational selection model) or solely induced upon binding to F24 TCR (induced fit model) we performed MD simulations of each HLA-DR molecule in their TCR-bound and free states (i.e. six systems in total) for 500 nanoseconds. No major variations in peptide binding cleft flexibility were observed over the course of the simulations (Appendix A). Regions α43–59, α72–80, and β17–26 displayed relatively large root mean square fluctuations (RMSF) (Figure 1C–F), which can be attributed to the movement of unstructured loop regions outside of the binding cleft, except region α43–59, which occupies the N-terminus of the HLA-DRα α-helix (Figure 1G,H and Appendix A).

Comparison of the RMSF distribution between HLA-DR-RQ13 and F24 TCR-HLA-DR-RQ13 structures, showed no significant differences in RMSF associated with the backbone atoms of the peptide binding cleft, except for a marginal increase in RMSF at HLA-DRβ cleft region (residues 50–70) which was seen in all F24-HLA-DR-RQ13 bound complexes when compared to free structures (Figure 1D–F). Overall, these results suggest that the polymorphic residues across all three HLA-DR allomorphs do not alter intrinsic flexibility, despite the open cleft conformation of HLA-DR1-RQ13 and HLA-DR15-RQ13. Thus, the closing of the peptide binding cleft is likely a consequence of F24 TCR binding, with minor regions of flexibility highlighting a level of plasticity within HLA-DR molecules required for TCR engagement.

### 2.2. Polymorphic Residue HLA-DRβ67 Dictates Peptide Binding Cleft Opening 

The MD RMS deviation (RMSD) plots (Appendix A) and ensemble refinement models (Appendix A) for HLA-DR-RQ13 and F24 TCR-HLA-DR-RQ13 indicated rigidity of secondary structures comprising the antigen binding cleft. We then used ensemble refinement to investigate side chain plasticity at the TCR-peptide-HLA interface and to understand TCR binding dynamics [13,18]. Our previous structural analysis suggested that HLA-DRβ polymorphic residues at positions 13, 26 and 67 (Appendix A) were key determinants of the different conformations of the cleft observed among the allomorphs [14]. To further understand the impact of HLA-DRβ polymorphism on TCR recognition, and establish if protein dynamics could reveal new mechanistic insights we used ensemble refinement. Ensemble refinement models showed very limited side chain variation in Phe26^β^ (HLA-DR11 and HLA-DR15) and Leu26^β^ (HLA-DR1) (Appendix A). Although Phe is larger than Leu, its aromatic side chain can adopt a planar conformation that packs neatly into the cleft. In contrast, the branched methyl side chain of Leu26^β^ contributes to HLA-DR1 having the largest cleft opening of all three HLA-DR-RQ13 structures. Another example of the favourable planar conformation of aromatic residue occurs in Phe67^β^ (HLA-DR11). The Phe67^β^ side chain is underneath the peptide’s glutamic acid residue at position 7 (P7-E, Figure 2A). In contrast, Ile67^β^ and Leu67^β^ (HLA-DR15 and HLA-DR1, respectively) and peptide P7-E side chains face each other and create less favourable interactions compared to Phe67^β^ (HLA-DR11) (Figure 2B,C).

In the HLA-DR-RQ13 free states, the P7-E side chain is exposed to the solvent (Figure 2A–C). The analysis of the relative B factor of the RQ13 side chains show that the solvent exposed residues have different mobility in the three HLA-DR allomorphs in their free state, without a clear allomorph being more or less flexible overall (Appendix A). The relative B factor analysis showed that P7-E side chain was the most mobile when bound to HLA-DR15 and less mobile when binding to HLA-DR1 (Appendix A). However, the extent of conformational flexibility of the peptide P7-E side chain, only accessible with ensemble refinement (Figure 2A–C), correlated with the size and side chain orientation of the HLA-DRβ67 polymorphic residue. Ensemble refinement showed that P7-E spatial variation was higher in HLA-DR1 and HLA-DR15 (standard deviation (sd) in RMSD of 0.044 Å) compare to HLA-DR11 (sdRMSD of 0.034 Å), and that globally the P7-E^free^ was mobile in all three HLA-DR allomorphs. In addition, the ensemble refinement also revealed that Phe67^β^ in HLA-DR11 was stable (sdRMSD of 0.008 Å), while Leu67^β^ and Ile67^β^ in HLA-DR1 and HLA-DR15 were more flexible (sdRMSD of 0.036 and 0.030 Å, respectively). This confirms that Phe67^β^ in HLA-DR11 is indeed well stabilised underneath the peptide P7-E residue (Figure 2A).

Upon binding to the F24 TCR, the peptide P7-E side chain adopts an upward perpendicular conformation, independent of the HLA-DR molecule, stabilised by hydrogen bonding with Ala110β from F24 CDR3β loop (Figure 2D,F) and associated with the closing of the cleft. Ensemble refinement (Appendix A) and relative B factors showed that the P7-E residue adopts a stable conformation in all three complexes (Appendix A). Interestingly, spatial variation of the P7-E side chain was also present in the F24 TCR-bound state of the HLA-DR-RQ13 complexes, albeit to a lesser degree than in the free state (Figure 2D,F). In addition, the differences in P7-E sdRMSD between free and bound states correlated with the F24 TCR affinities. The largest difference between free and bound states was observed for HLA-DR1 (sdRMSD of 0.033 Å), which has the lowest affinity (Kd of 10.56 µM) [14]; then for HLA-DR15 (sdRMSD of 0.017 Å) with the intermediate affinity (Kd of 6.90 µM) [14]; and finally for HLA-DR11 (sdRMSD of 0.005 Å), which has the highest affinity (Kd of 1.16 µM) [14]. We also observed changes in sdRMSD for the HLA–DR67β residue upon F24 TCR binding, although not to the same extent as for the P7-E residue (sdRMSD variation between free and bound states range: 0.002–0.008 Å). As such, in the HLA-DR-RQ13 bound, the Phe67^β^ and P7-E pairs (HLA-DR11) showed limited spatial variation (Figure 2G), Ile67^β^ and P7-E showed moderate spatial variation (HLA-DR15, Figure 2H), and Leu67^β^ and P7-E showed the highest spatial variation (HLA-DR1, Figure 2I). While the P7-E side chain was stable across all bound states of the HLA-DR allomorphs, the MD analysis showed that HLA-DR11-RQ13^bound^ was able to adopt different conformations, one resulting in an upward shift of the HLA-DRβ cleft and closer hydrogen bond between P7-E and Ala110β, which was not observed for HLA-DR1 or HLA-DR15 (Appendix A). These results suggest that the interaction between RQ13 P7-E and F24 TCR Ala110β is a key driver that reshapes the HLA-DR antigen binding cleft upon TCR binding.

Taken together, these analyses imply that conformational dynamics of the HLA-DRβ67 polymorphism not only plays a role in dictating the opening of the cleft but also facilitates the interaction between P7-E and Ala110β during TCR binding. Ensemble refinement analysis reveals that the F24 TCR samples the available P7-E conformations, and therefore that conformational selection mechanism underpins F24 TCR recognition.

### 2.3. HLA-DR Polymorphisms Destabilise an Intricate TCR Peg-Notch Interaction

The F24 TCR interacts via its CDR3β loop by a peg-notch interaction, wherein the Met113β side chain is inserted into a notch formed by the polymorphic residues HLA-DRβ67, -DRβ70 and -DRβ71 (Appendix A). The HLA-DRβ67 interaction with P7-E and Ala110β (discussed above) forms the C-terminal wall of the notch, whilst interactions between HLA-DR70β, 71β and P5-R form the N-terminal wall of the notch (Appendix A). Ensemble refinement analysis reveals that whilst this peg-notch interaction is stable in the complex with HLA-DR11 (Figure 3A), it is conformationally heterogenous in HLA-DR15 and HLA-DR1 (Figure 3B,C). Indeed, the B factor analysis of the CDR3β loop in the three F24 TCR-HLA-DR-RQ13 complexes revealed that the main chain of Met113β has the highest relative B factor compared to other residues (Appendix A). In addition, the Met113β side chain relative B factor was higher in the HLA-DR15 and HLA-DR1 complexes than HLA-DR11 (Appendix A), where the residue is stabilised in a peg-notch. In HLA-DR11 (stable), the polymorphic residues Asp70^β^ and Arg71^β^ within the HLA-DRβ cleft play a role in stabilising each other, via a salt bridge, and Arg71^β^ forms hydrogen bonds with the carbonyl group of P5-R. These interactions “lock in” the structural orientation of Asp70^β^ and Arg71^β^ side chains, and together with P5-R, form the N-terminal wall of the notch in which the F24 TCR Met113β pegs into (Figure 3A). In HLA-DR15 (Ala71^β^ and Gln70^β^), the short Ala71^β^ side chain cannot interact with Gln70^β^ or P-5R and allows for spatial variation of Gln70^β^ side chain orientation (Figure 3B). This leads to an opening of the N-terminal side of the notch and destabilises the Met113β peg-notch interaction as represented by the diverse possible structural orientations of Met113β side chain observed in ensemble refinement of F24 TCR-HLA-DR15-RQ13 complex (Figure 3B). Interestingly, the crystal structure shows that the Gln70^β^ side chain can form a hydrogen bond to P5-R carbonyl; however, the ensemble models suggest that this interaction is transient and still allows for an opening within the notch. In HLA-DR1 (Gln70^β^ and Arg71^β^), the crystal structure show that Arg71^β^ hydrogen bonds with P5-R carbonyl, but not the Gln70^β^ side chain (Figure 3C), yet the ensemble models show that this interaction could exist similarly to the one observed in HLA-DR15 complex (Figure 3B). However, due to a larger open cleft conformation of HLA-DR1 and structural variation of P7-E, the notch is not constrained and allows for a larger flexibility of the F24 TCR Met113β side chain (Figure 3C). Thus, ensemble refinement models of F24 TCR-HLA-DR1-RQ13 show a high degree of variation in position and orientation for F24 TCR Met113β when compared to F24 TCR-HLA-DR11-RQ13 (Figure 3A).

These observations are consistent with mutagenesis studies that measured binding affinities of F24 TCR with HLA-DR-RQ13 molecules, such that an alanine mutation of F24 TCR Met113β led to >20-fold loss in binding affinity for HLA-DR11, 11-fold loss for HLA-DR15 and 9-fold loss for HLA-DR1 [14]. Ensemble refinement, revealing local structural variation of key residue side chains such as Met113β, provides a rationale for these binding data, suggesting a fine specificity in the F24 TCR recognition for the different HLA-DR allomorphs. Additionally, we showed that alanine mutation of HLA-DR11 Asp66^β^ increases F24 TCR binding affinity by 3-fold for HLA-DR11-RQ13, where the binding kinetics were also impacted by a slower off rate [14]. The ensemble refinement is consistent with these observations, based on the molecular interactions of Asp66^β^ (Figure 3D), where its side chain is constrained by and changes the structural orientation of the CDR3β loop (between Met113β, Asp117β and Glu118β). The mutation of Asp66^β^ into Ala66^β^ would minimise steric clashes with the CDR3β loops and therefore change the kinetics of binding. We therefore hypothesised that the Ala66^β^ mutation in HLA-DR1 would also improve the F24 TCR affinity and change its kinetics. To test this hypothesis, we determined that the F24 TCR affinity for the HLA-DR1-D66A^β^ mutant in complex with RQ13 peptide (Kd of 5.5 μM) showed a minor (2-fold) increase compared to HLA-DR1-RQ13 (Kd of 10.56 μM [14]). However, consistent with the HLA-DR11-D66A^β^ mutation, we also observed a slower off-rate when binding to its wild-type counterpart [14] (Appendix A). These results indicate that Asp66^β^ impairs the ability of the CDR3β loop to form interactions that increase TCR-pHLA affinity in both HLA-DR1 and HLA-DR11 molecules. This was reflected by the observed slower dissociation rates, and higher F24 TCR affinity towards Ala66^β^ mutant compared to wild type.

Comparison of the F24 TCR structures in bound and free states showed that the CDR3β loop changed conformation upon TCR binding to avoid steric clashes with HLA-DRβ α-helix, with the greatest conformational changes observed for F24 TCR Met113β and Ala110β [14]. Our ensemble refinement analysis showed that the change in conformation for Ala110β is likely due to its interaction with P7-E, stabilised by the polymorphic residue HLA-DRβ67 (Figure 2D,F). In comparison, the conformational change within Met113β is primarily stabilised by the formation of the peg-notch interaction (Figure 3A–C). These two interactions therefore dictate the CDR3β loop conformational change required for F24 TCR recognition and binding of HLA-DR-RQ13 molecules. Indeed, ensemble refinement models of F24 TCR free shows that Met113β has the greatest structural deviation of all residues within the CDR3β loop (Figure 3E), some being favourable with the binding of the HLA-DR-RQ13 complex, supporting a conformational selection model of interaction. This also suggests that the polymorphic residues in HLA-DRβ that dictate the shape of the cleft and are involved in stabilising the peg-notch interaction, can impact TCR affinity.

### 2.4. Structural Rigidity Is a Shared Feature of pHLA-DR Complexes

To understand whether the observed structural rigidity of HLA-DR-RQ13 binding clefts in MD simulations and ensemble refinement models was specific to RQ13 bound complexes, we extended the ensemble refinement analysis to 41 pHLA-DR structures that fit into our selection criteria (see Methods, Table 1). Ensemble refinement RMSF values for HLA-DRβ peptide binding cleft (residues 7–90) of HLA-DR1, HLA-DR11 and HLA-DR15 structures bound to RQ13 peptide (RMSF mean of 0.40, 0.38, and 0.41, respectively) were within the range of the HLA-DRβ peptide binding cleft of molecules bound to other peptides (RMSF mean ± SD = 0.47 ± 0.29). The mean RMSF of bound peptide residues (P1–P9) was also measured, revealing that RQ13 binding to HLA-DR1, HLA-DR11 and HLA-DR15 (RMSF mean of 0.37, 0.21 and 0.58, respectively) falls in the same range as other pHLA-DR complexes (RMSF mean ± SD of 0.44 ± 0.24). These results suggest that the structural rigidity observed in the HLA-DR-RQ13 complexes is a shared feature of peptide-HLA-DR complexes.

In total, 12 HLA-DR allomorphs structures are available (Figure 4), however only 7 HLA-DR1 (HLA-DRB1 *01:01) and 12 HLA-DR4 (HLA-DRB1 *04:01) structures have been solved with multiple peptides (with a resolution cut-off of <2.5 Å). Therefore, we focused on these two HLA-DR molecules to observe whether there was a difference between HLA-DR allomorphs. Once again, we observed that there was no significant difference between the RMSF of the HLA-DRβ peptide binding cleft (RMSF mean ± SD of 0.61 ± 0.32 and 0.32 ± 0.12 for β-chain, respectively) or bound peptide residues (RMSF mean ± SD of 0.45 ± 0.20 and 0.494 ± 0.35, respectively), suggesting that there is no difference in binding cleft flexibility between those HLA-DR allomorphs. Despite a similar rigidity of the antigen binding cleft, the HLA-DR1 could adopt a boarder range of cleft conformations, with a maximum displacement of 1.4 Å (Cα of residue HLA-DR64β), compared to 0.6 Å for HLA-DR4 (Figure 4A,B). With the exception of HLA–DRB5 *01:01, for which only two structures are available to date, HLA-DR1 seems to present the largest range of different cleft conformations, and this may impact on T cell recognition and peptide binding.

### 2.5. HLA-DR1 Displays High Stability and Low HLA-DM Susceptibility Despite Its Open Cleft Conformation

We have shown that the open-cleft conformation caused by polymorphic residues does not lead to increased flexibility (Figure 1), especially in HLA-DR1-RQ13 vs HLA-DR11-RQ13, and that it is a shared feature in other HLA-DR molecules. However, it is unclear whether the open conformation of the cleft could allow for a faster peptide exchange or lower stability of the overall pHLA complex, as these parameters are important factors for T cell immunity and TCR binding [39,40] We focused on HLA-DR1 and HLA-DR11 as they have the most extreme conformations of the antigen binding cleft when bound to RQ13 peptide. We performed thermal shift assays to assess the stability of HLA-DR1 and HLA-DR11 in complex with CLIP, RQ13 peptide, or the well-studied HA_306–318_ influenza peptide [41].

Surprisingly, the HLA-DR11 (closed cleft) and HLA-DR1 (open cleft) in complex with either the RQ13 or HA_306–318_ peptides showed a difference in protein stability with a thermal melting temperature (T_m_), of 81 °C for pHLA-DR1 and 60–64 °C for pHLA-DR11 (Table 2). We expected both HLA-DRs in complex with CLIP to show a substantial decrease in T_m_ (Table 2) as CLIP is a self-peptide that stabilises the HLA-II dimer in the endoplasmic reticulum, that will be cleaved and replaced by a high affinity antigenic peptide with the help of HLA-DM [42,43] While the stability of HLA-DR11-CLIP was low, with a T_m_ of 46 °C, the HLA-DR1-CLIP stability was higher than expected with a T_m_ of 71 °C (Table 2). The T_m_ of HLA-DR1-CLIP is on average 10 °C higher than classical HLA class I complexes such as HLA-A2 presenting the M1 influenza peptide (T_m_ of 59 °C) (Table 2). Seemingly, this suggests that regardless of the bound peptide, HLA–DR1 is more stable than HLA-DR11.

As CLIP represents a low affinity peptide for all MHC class II molecules, the reason for such a large difference in the overall protein stability between HLA-DR11-CLIP and HLA-DR1-CLIP is not understood. The residues forming the P1 pocket in HLA-DR1 and HLA-DR11 are fully conserved, and therefore the difference might be attributable to the P4 pocket. The CLIP peptide has a small alanine residue at P-4, which in HLA-DR1 interacts with Phe13 at the base of the cleft, but the smaller Ser13 of HLA-DR11 is more than 4 Å apart from P4-A and might not be able to stabilise the CLIP peptide as efficiently (Appendix A). Regardless, HLA-DR1’s observed open cleft does not reduce the stability of the overall pHLA complex, and HLA-DR1 is inherently stable with the CLIP peptide.

While the open cleft conformation of HLA-DR1 did not reduce the overall stability of the pHLA complex, it is unknown whether it could alter peptide exchange by HLA-DM, or peptide affinity relative to HLA-DR11. HLA-DM is known to bind and stabilise MHC-II during peptide loading [42,43]. Interestingly, while our MD analysis did not show a significant difference between the HLA-DR-RQ13 free and F24 TCR-bound states, some localised differences were observed. The area around HLA-DRα55 reached an RMSF (mean ± SD) of 1.7 ± 0.2 Å in HLA-DR11-RQ13 free, 1.5 ± 0.5 Å in HLA-DR15-RQ13 free, that was decreased to 0.8 ± 0.1 Å and 0.8 ± 0.02 Å, respectively, in both F24 TCR-bound complexes (Figure 1G,H). Contrastingly, HLA-DR1 showed a modest reduction in RMSF of 1.0 ± 0.1 Å and 0.7 ± 0.1 Å between free and TCR-bound states, respectively (Figure 1G,H). The region surrounding HLA-DRα55 is involved in both the stabilisation of N-terminal peptide (P-2, P-1) [4] and HLA-DM stabilisation of the P1 pocket ^6^. Upon binding to HLA-DM, the unstructured loop in HLA-DRα (residues 52–55) folds into an elongated α-helix, which stabilises the P1 pocket during peptide exchange [6]. This region may possess intrinsic flexibility that allows for the dual functionality of peptide stabilisation and HLA-DM interaction during peptide loading.

As HLA-DR1 shows less flexibility in this region of the HLA-DRα chain, and exhibits higher protein stability, this suggests that HLA-DR1 may be less susceptible to HLA-DM than HLA-DR11. To test this, we performed a peptide exchange assay measured by fluorescence polarisation, with HLA-DR11 or HLA-DR1 in the presence or absence of HLA-DM and increasing concentrations of RQ13 (Appendix A and Table 3). Our results showed that in the absence of HLA-DM both HLA-DR11 and HLA-DR1 had similar affinity for RQ13 peptide with an IC_50_ of 161 nM and 195 nM, respectively (Table 3). Surprisingly, while HLA-DR11 showed a 5-fold higher affinity for RQ13 in the presence of HLA-DM (IC_50_ of 32 nM), no significant change was observed for HLA-DR1 affinity for RQ13 in the presence of HLA-DM (IC_50_ of 290 nM). This data shows that HLA-DR1 is less HLA-DM susceptible than HLA-DR11.

## 3. Discussion

In this study, we have used crystallographic ensemble refinement to investigate the structural flexibility of HLA-II molecules that could impact peptide presentation, stability of the pHLA-II complex as well as T cell recognition. We firstly focused on three HLA-DR molecules binding the same peptide, and then expanded our analysis to 41 pHLA-DR complexes, for which structures were available. MD analysis was able to reveal regions of high flexibility; however, there was no significant difference between the three HLA-DR-RQ13 complexes. Although ensemble refinement shows a lack of conformational variability and provides some evidence against substantial conformational flexibility, it does not necessarily imply that the structure is always rigid, but suggests that there is a lack of structural variation in the crystal form in which the ensemble is calculated. Nevertheless, our results reveal a rigid antigen binding cleft for all three HLA-DR-RQ13 complexes, suggesting that F24 TCR binds with a conformational selection model rather than through intrinsic flexibility of the HLA-DR binding cleft.

The level of rigidity in MHC-II could be linked with the high stability associated with MHC-II over MHC-I [10,11]; however, this was surprising given the high number of polymorphic residues in the cleft among the allomorphs that could affect flexibility. This prompted us to investigate specific polymorphic residues and their interactions to provide a basis for the different cleft conformations and binding affinities observed for the F24 TCR [14]. Ensemble refinement allowed us to analyse the spatial variation of polymorphic residue side chains and unveiled an interaction between HLA-DRβ67 and P7-E. The size and orientation of HLA-DRβ67 side chains dictated the cleft opening within each HLA-DR-RQ13 complex. Ensemble refinement analysis of TCR-bound complexes showed that the closing of the cleft was associated with a displacement of the P7-E side chain favoured by the formation of a hydrogen bond with the F24 TCR CDR3β loop. HLA-DRβ67 and P7-E are also involved in the stabilisation of a peg-notch interaction where the F24 TCR Met113β forms one side of the notch, whilst polymorphic residues HLA-DRβ70–71 and P5-E form the other side. The disruption of any interactions involving the side chains of polymorphic residues 67, 70 or 71 of HLA-DRβ, ultimately led to the destabilisation of P7-E/Ala110β interaction and the Met113β peg-notch. The extent of these disruptions correlated with the hierarchy of F24 TCR affinity observed among HLA-DR-RQ13 complexes [14].

The open binding cleft does not cause the loss of affinity to the F24 TCR directly, but rather indirectly affects the stable interactions (peg-notch) required for high affinity TCR binding. This is an interesting and unique case, in which the binding of the RQ13 peptide can change the shape of the HLA-II binding cleft due to the polymorphic residues located in the cleft. It is well understood that polymorphic residues at the peptide binding cleft within each HLA allotype dictate the peptide repertoire; in this case of cross-presentation. Our study suggests that both the HLA polymorphic residues and the peptide residues can play a role in moulding the antigen binding cleft, impact on TCR affinity and recognition, and potentially influence T cell recognition.

To find out whether this rigidity was a unique feature of the RQ13 peptide, we performed ensemble refinement on the 41 pHLA-DR crystal structures available. We found that rigidity of the cleft was a common trait within all pHLA–DR structures, and in great contrast to the structural plasticity of MHC-I [13]. This was surprising, as superimposition of all structures showed significant variation in the size of the antigen binding cleft, and so one could expect that some HLA-DRs may be more flexible than others. This led us to speculate whether an open cleft conformation could affect protein stability or HLA-DM susceptibility, as previous studies have showed that these two determinants can correlate with immunodominance [39,40]. Surprisingly, HLA-DR1 possessed inherently higher stability than HLA-DR11 when presenting the RQ13 and HA_306–318_ peptides, and even the CLIP peptide. Stability has been found to be a better predictor of immunogenicity than peptide affinity for MHC class I [12,39], which in the light of the high stability of HLA-DR1-CLIP may not be the case for HLA-DR1 or for MHC-II molecules in general.

At the cellular level, HLA-DR molecules are expressed with CLIP in the endoplasmic reticulum, as HLA-II are unstable with a vacant binding cleft. In late-endosomal compartments, CLIP is cleaved and its dissociation is induced by HLA-DM, which also serves to stabilise the empty binding cleft. The process of peptide exchange is efficient because high affinity peptides must bind to HLA-II and ultimately compete with HLA-DM for successful peptide loading [44]. Given the HLA-DR1-CLIP high stability, successful loading of peptides onto the binding cleft of HLA-DR1 should be facilitated by HLA-DM, ensuring that only the highest affinity peptides are presented. However, our peptide affinity in the presence and absence of HLA-DM showed that HLA-DR1’s affinity for RQ13 peptide was unchanged (IC_50_ of ~200–300 nM). On the other hand, HLA-DR11 showed a 5-fold increased affinity for RQ13 in the presence of HLA-DM.

Generally, pHLA-DR complexes that are known to have high HLA-DM susceptibility are indicative of peptide immunogenicity [40,44]. In addition, HLA-II allomorphs that bind poorly to HLA-DM, such as HLA-DQ2, are associated with increased risk for autoimmune diseases such as type 1 diabetes and coeliac disease [45,46]. Here, we showed that the same immunogenic RQ13 peptide presented by both HLA-DR1 and HLA-DR11 can give rise to different HLA-DM susceptibility and different peptide affinity. We suggest therefore that the link with immunogenicity is not solely based on those parameters. In addition, it has been shown previously that the HLA-DR1-HA_306–318_ complex has a low susceptibility to HL-DM [4,40,47,48,49,50,51,52], despite HA_306–318_ being an immunodominant epitope. Interestingly, slower dissociation rates between MHC-II and peptide is a trait associated with low HLA-DM susceptibility (reviewed in [53]). This could mean that despite an open cleft conformation, the high stability of HLA-DR1-RQ13 could contribute to a long-lived, stable pHLA-II complex presented on the cell surface.

These results provide a combinatorial approach for understanding the structural features of antigen presentation by HLA-II molecules and recognition by T cells. Here, we revealed how localised residue flexibility within the HLA-II or peptide directly impacts TCR affinity, suggesting that an intricate interplay of structural rearrangement during TCR binding drives TCR affinity, which is known to shape T cell activation [14]. Interestingly, despite HLA-DR1’s open cleft conformation, it was associated with high pHLA stability even when presenting the CLIP peptide. This might suggest that the HLA-DR1 peptide repertoire is different from other HLA-DR molecules, such as HLA-DR11, allowing for either a larger number or more diverse peptide repertoire to be presented. Taken together with the lower HLA-DM susceptibility, this suggests that HLA-DR1 might be a stable and versatile presenter of peptides in an environment where antigen presentation is vital to mounting an effective immune response.

## 4. Methods

### 4.1. HLA-DR-CLIP Expression, Purification, and Peptide Loading

HLA-DR-CLIP molecules were expressed using adherent Human embryonic kidney (HEK) 293S cells (ATCC# CRL-3022, GnTI^-^) [14]. HEK293S cells were transiently co-transfected with pHLsec vector containing either the HLA-DRβ-CLIP and HLA-DRα using polyethylenimine (PEI). The cells were incubated at 37 °C, 5%CO_2_ over the course of 6–10 days. The HLA-DR-CLIP protein was then harvested from the cell culture and purified from the expression media. Firstly, incubation for 30 mins with 50 mM Tris-HCl pH 8, 1 mM NiCl_2_, and 5 mM CaCl_2_. The supernatant was centrifuged and filtered before applying to a GE 5 mL HisTrap column, eluted with 10 mM Tris-HCl pH 8.0, 150 mM NaCl, 300 mM Imidazole pH 8. The HLA-DR-CLIP proteins were then further purified and buffer exchanged by GE S200 16/600 gel filtration. The CLIP peptide was exchanged with the RQ13 or HA_306–318_ in the presence of HLA-DM [14]. Once the peptide exchange has been facilitated via incubation for 12 h at 37 °C, at pH 5.4 in Citrate buffer, the loaded HLA-DR-RQ13 or HLA-DR-HA_306–318_ were further purified via anion exchange liquid chromatography.

### 4.2. F24 TCR Production and Surface Plasmon Resonance

The F24 TCR was prepared as previously described [14] to be used for surface plasmon resonance (SPR). Briefly, F24 TCRα and TCRβ subunits were expressed as inclusion bodies in E. *coli* cells, before being refolded in 5 M Urea, 0.1 M Tris-HCl pH 8.0, 2 mM Ethylenediaminetetraacetic acid (EDTA) pH 8, 10 mM glutathione (reduced), 2.5 mM glutathione (oxidized) for 72 h. The refold mixture was dialysed in 10 mM Tris-HCl pH 8.0 and purified using diethylaminoethyl cellulose anion exchange, hydrophobic interaction chromatography (HiTrap HIC) and finally anion exchange (HiTrap Q) again. SPR experiments were conducted on BIAcore T100 (GE Healthcare, Chicago, IL, USA) at 25 °C in 10 mM Tris-HCl pH 8, 150 mM NaCl, 0.005% Tween-20, and 1% BSA. Biotin-tagged HLA-DR1-RQ13 complex was coupled onto a BIAcore Sensor Chip SA (GE Healthcare, Chicago, IL, USA) to ~2000 response units. HLA-DR1-CLIP was used as a control on the reference flow cell. SPR experiments were performed at 10 µL/min with ten serial dilutions at 200 µM for F24 TCR. BIAevaluation Version 3.1 (GE Healthcare, Chicago, IL, USA) and GraphPad Prism 8 (GraphPad, San Diego, CA, USA) was used to analyse and plot the sensorgrams, respectively.

### 4.3. Thermal Stability Assay

To assess the stability of each pHLA–DR complex, we performed a thermal-shift (or thermal denaturation) assay. The fluorescent dye SYPRO^™^ Orange (Invitrogen, Carlsbad, CA, USA) was used to monitor the protein unfolding. Real-time fluorescence detection was conducted using Rotor-Gene Q (QIAGEN, Hilden, Germany). Each pHLA-DR complex was in 10 mM Tris-HCl pH 8.0, 150 mM NaCl, at two concentrations (0.8 and 1.6 µM) in duplicate and was heated from 25 to 95 °C at a rate of 0.5 °C/min. The fluorescence intensity was measured with excitation at 530 nm and emission at 557 nm. The data collected was processed and presented using GraphPad Prism 8 (GraphPad, San Diego, CA, USA). A sigmoidal non-linear regression was performed. From the generated curve, the melting temperature (T_m_) was calculated, which represents the midpoint in the unfolding process (the temperature where 50% of the protein is unfolded).

### 4.4. Fluorescence Polarization Assay

To determine the relative affinity of the RQ13 peptide for HLA-DR1 or HLA-DR11 with and without DM, we performed a fluorescence polarization assay as previously described [54,55]. In brief, 20 nM Alexa Fluor 488-PRFVKQNTLRLAT peptide, 100 nM HLA-DR1 or HLA-DR11 with or without 20 nM HLA-DM was incubated in wells of a five-fold dilution series of the RQ13 peptide (from 250 μM–0 μM) in Citrate buffer pH 5.4. The plate was incubated at 37 °C for 24 h. The fluorescence polarization was then measured for each well using the BMG Pherastar FS plate reader (BMG Labtech, Offenburg, Germany). Data were processed using Graphpad Prism 8 version 8.4.2 where the values were converted to the percentage bound using the equation (FP_sample_ – FP_free_)/(FP_no comp_ – FP_free_); where FP_sample_ represents the FP detected for wells containing the competitor peptide; FP_free_ is the background fluorescence of the HA-TAMRA peptide unbound to HLA, and the FP_no comp_ are the FP values for wells without competitor peptide. A sigmoidal dose–response model was fitted to determine the half maximal inhibitory concentrations (IC_50_).

### 4.5. Statistical Analysis for Fluorescence Polarization Assay

Statistics were computed using GraphPad Prism 8 (GraphPad, San Diego, CA, USA). *p*-values <0.05 were considered statistically significant. Differences between groups were analysed with one-way analysis of variance (ANOVA) followed by a Bonferroni multiple comparisons post-hoc test. Half maximal inhibitory concentrations (IC_50_) were obtained after a sigmoidal dose-response model was fitted. All significant differences between the groups (*p* < 0.05) are reported in the data plots.

### 4.6. Computational Resources

Ensemble Refinement and Molecular Dynamics Simulations were performed on in house hardware (NVIDIA TITAN X Pascal GPU, Santa Clara, CA, USA).

### 4.7. Ensemble Refinement

We identified 43 pHLA-DR structures from the PDB (San Diego, CA, USA); 2 of them were excluded as the reflection data (mtz file) was not available (1DLH, 1BX2). All atomic coordinate (.pdb) and crystallographic reflection (.mtz) files were sourced from the PDB_REDO server (San Diego, CA, USA) [56]. Ensembles were calculated with PHENIX 1.9–1692 (Berkeley, CA, USA) [57] by first passing each system through the ReadySet tool (Berkeley, CA, USA) and then using the Ensemble Refinement tool (Berkeley, CA, USA)with default parameters [17]. Ensemble analysis was performed using PyMOL version 1.3 (Schrodinger©, New York, NY, USA) and Python scripts (Monash University, Clayton, Australia), whilst Mean RMSF ± SD calculations were analysed using GraphPad Prism 8 (GraphPad, San Diego, CA, USA), where ensemble models with a crystal structure resolution above 2.5Å were omitted from calculations.

### 4.8. Atomic Coordinates, Modelling, and Graphics

In MD simulations, atomic coordinates were obtained from the following PDB entries: 6CPN, 6CQJ, 6CPO, 6CQQ, 6CQL, 6CQR [14]. Structural representations were produced using PyMOL version 1.3 (Schrodinger ©, New York, NY, USA) and VMD 1.9.1 (Urbana-Champaign, IL, USA) [58]. Trajectory manipulation and analysis was performed using MDTraj (Stanford, CA, USA) [59] and VMD 1.9.1.

### 4.9. MD Systems Setup and Simulation

Each protein, with protonation states appropriate for pH 7.0 [60] was placed in a rectangular box with a border of at least 12Å and explicitly solvated with TIP3P water [61]. Counterions were added, and the proteins were parameterized using the AMBER ff14SB all-atom force field [62]. After an energy minimization stage and an equilibration stage, production simulations were performed in the isothermal–isobaric NPT ensemble (300 K, 1 atm) via the use of a Berendsen barostat and Langevin thermostat with a damping coefficient of 2 ps^−1^. Three independent replicates of each system were simulated for 500 ns each using Amber18 [63]. Energy minimization and equilibration stages consisted of energy minimization via steepest descent followed by conjugate gradient descent until convergence, followed by a heat gradient over a 1 ns period with positional restraints on all non-water molecules, followed by a 0.2 ns density revision step using isotropic pressure scaling with the same position restraints. The system was then allowed to equilibrate without restraints for 7 ns. Further information regarding the protocols can be found inside the stageUtil package https://github.com/blake-riley/MD_stageUtil/tree/master/_init/c-Staging/_templates (open source, Monash University, Clayton, Australia).

## Figures and Tables

**Figure 1 ijms-21-07081-f001:**
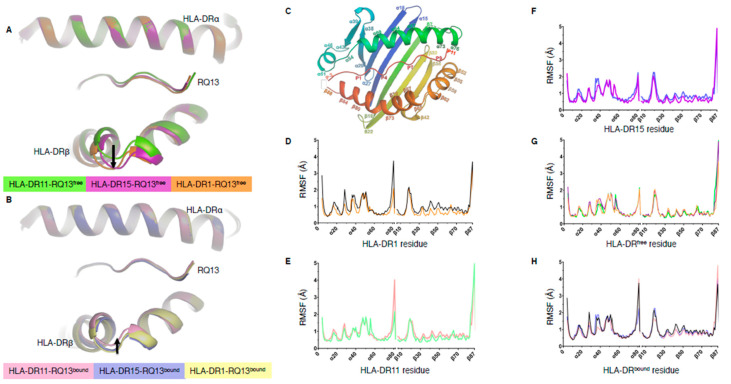
Structural alignment of HLA-DRβ cleft and MD RMSF distributions. (**A**) Structural alignment of HLA-DR11 (Protein Data Bank (PDB) ID: 6CPN, green), HLA-DR15 (PDB ID: 6CPO, pink), HLA-DR1 (PDB ID: 6CQJ, orange) in complex with RQ13 (free state) and (**B**) F24 TCR-bound state (PDB ID: 6CQL in pale pink, 6CQQ in pale purple, 6CQR in yellow, respectively). The arrows indicate the opening (**A**) or closing (**B**) of the HLA-DR antigen binding cleft. (**C**) HLA-II structure showing amino acid numbers on the antigen binding cleft. (**D**–**F**) RMSF distribution between free and F24 TCR-bound states of (**D**) HLA-DR1 (orange and black, respectively), (**E**) HLA-DR11 (green and pale pink, respectively), and (**F**) HLA-DR15 (pink and pale purple, respectively). (**G**) RSMF distribution of all HLA-DR-RQ13 complexes in the free state. (**H**) RMSF distribution of all HLA-DR-RQ13 complexes in the F24 TCR-bound state.

**Figure 2 ijms-21-07081-f002:**
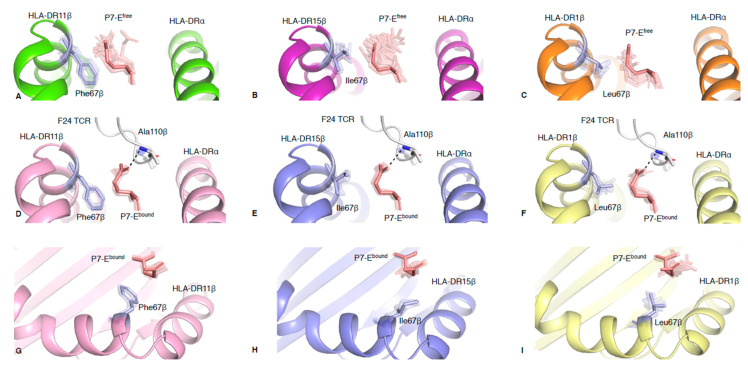
Conformational populations between side chain interactions of HLA-DRβ67 and P7-E. (**A**–**C**) Ensemble results of HLA-DRβ67 side chain (pale blue) interacting with P7-E residue (pale red) in the free state for (**A**) HLA-DR11-RQ13, (**B**) HLA-DR15-RQ13, (**C**) HLA-DR1-RQ13. (**D**–**I**) Ensemble results for F24 TCR-bound state of P7-E side chain (pale red) interacting with F24 TCR Ala110β (white) and HLA-DRβ67 side chain (pale blue) for (**D**) HLA-DR11-RQ13 (**E**) HLA-DR15-RQ13 (**F**) HLA-DR1-RQ13. (**D**,**F**) the dashed lines show the hydrogen bonds between the P7-E and Ala110β. (**G**–**I**) Top view of ensemble results for F24 TCR-bound state of HLA-DRβ67 side chain and P7-E residue of (**G**) HLA-DR11-RQ13, (**H**) HLA-DR15-RQ13, and (**I**) HLA-DR1-RQ13. The crystal structure conformation is represented as solid colour, and the ensemble models are represented in transparent, both with the same colour coding.

**Figure 3 ijms-21-07081-f003:**
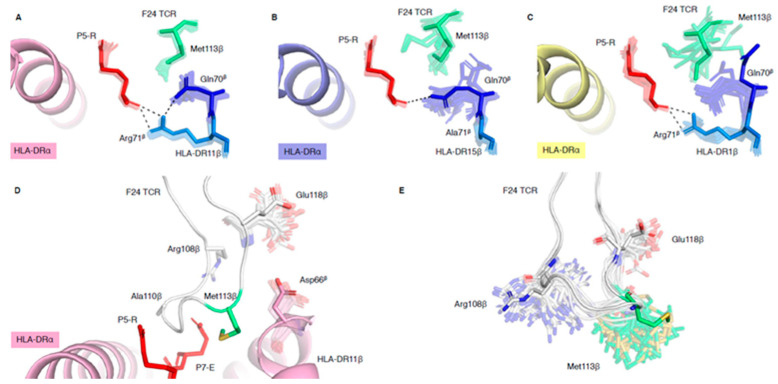
Spatial variation of side chains that form the F24 TCR Met113β peg-notch interaction. (**A**) Ensemble results of a stable peg-notch interaction in HLA-DR11-RQ13 (pale pink HLA-DRα), and disrupted peg-notch interaction, due to conformational variation of polymorphic residues, in (**B**) HLA-DR15-RQ13 (pale purple HLA-DRα), and (**C**) HLA-DR1-RQ13 (yellow HLA-DRα). These side chain interactions (dash lines) between HLA-DRβ70 (blue), HLA-DRβ71 (light blue) and P5-R (red) destabilise Met113β binding (green). (**D**) Ensemble results showing steric clashes between HLA-DR11 Asp66β (pale pink) and F24 TCR Glu118β (white) that affect binding. (**E**) Ensemble results of CDR3β loop residues derived from the crystal structure of F24 TCR free (PDB ID: 6CPH) showing a high level of conformational variation for Met113β, Arg108β and Glu118β.

**Figure 4 ijms-21-07081-f004:**
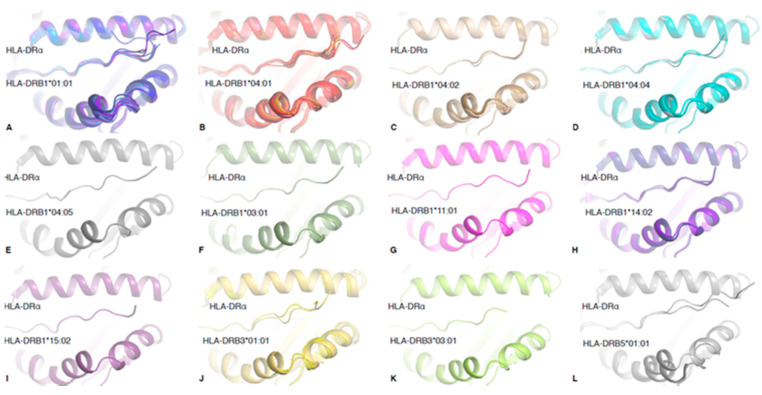
Overview of the available structures for pHLA-DR allomorphs. Crystal structures of the pHLA-DR complexes used for our analysis and reported in Table 1. Each structure of the same HLA-DR allomorph has been superimposed, and all pHLA-DR complexes are represented in the same orientation. The 12 available allomorph structures available are represented in (**A**) blue for HLA-DRB1 *01:01, (**B**) red for HLA-DRB1*04:01, (**C**) beige for HLA-DRB1*04:02, (**D**) cyan for HLA-DRB1 *04:04, (**E**) grey for HLA-DRB1*04:05, (**F**) green for HLA-DRB1*03:01, (**G**) pink for HLA-DRB1 *11:01, (**H**) bright purple for HLA-DRB1 *14:02, (**I**) dark purple for HLA-DRB1 *15:02, (**J**) yellow for HLA-DRB3*01:01, (**K**) lime for HLA-DRB3*03:01, and (**L**) grey for HLA-DRB5*01:01.

**Table 1 ijms-21-07081-t001:** pHLA-DR structures investigated by ensemble refinement.

PDB	Allomorph	Resolution (Å)	Peptide Sequence	Peptide Name	R_free_ (PDB)	R_free_ (Refine)	R_free_ (Ensemble)	ΔR_Free_	Ensemble Size	Reference
3L6F	HLA-DRB1*01:01	2.10	APPAYEKL(SEP)AEQSPP	MART-1	0.249	0.311	0.214	0.098	56	[19]
3PDO	HLA-DRB1*01:01	1.95	KPVSKMRMATPLLMQALPM	CLIP_102–120_	0.240	0.417	0.225	0.192	50	[20]
3QXA	HLA-DRB1*01:01	2.71	KPVSKMRMATPLLMQALPM	CLIP_102–120_	0.235	0.411	0.238	0.173	30	[21]
4I5B	HLA-DRB1*01:01	2.12	VVKQNCLKLATK	HA_308–319_	0.237	0.329	0.215	0.114	45	[22]
4OV5	HLA-DRB1*01:01	2.20	GSDARFLRGYHLYA	HLA-A2_104–117_	0.239	0.327	0.241	0.002	34	[23]
4X5W	HLA-DRB1*01:01	1.34	KPVSKWRMATPLLMQALPM	CLIP_102–120-W107_	0.170	0.210	0.166	0.004	110	[24]
6CQJ	HLA-DRB1*01:01	2.75	RFYKTLRAEQASQ	HIV_RQ13_	0.249	0.376	0.268	0.108	30	[14]
6HBY	HLA-DRB1*01:01	1.95	ARRPPLAELAALNLSGSRL	5T4 tumour	0.241	0.427	0.258	0.169	42	[25]
1T5W	HLA-DRB1*01:01	2.40	AAYSDQATPLLLSPR	MIG1_448–460_	0.255	0.356	0.229	0.127	39	[26]
1A6A	HLA-DRB1*03:01	2.75	KPVSKMRMATPLLMQALPM	CLIP_102–120_	0.325	0.372	0.297	0.028	38	[27]
4IS6	HLA-DRB1*04:01	2.50	WNRQLYPEWTEAQRLD	GP100	0.298	0.400	0.269	0.131	30	[28]
4MCY	HLA-DRB1*04:01	2.30	SAVRL-CIT-SSVPGVR	Vimentin_66–78-Cit_	0.225	0.314	0.235	0.079	30	[29]
4MCZ	HLA-DRB1*04:01	2.41	GVYAT-CIT-SSAVRLR	Vimentin_59–71-Cit_	0.231	0.407	0.236	0.005	24	[29]
4MD0	HLA-DRB1*04:01	2.19	GVYAT-CIT-SSAV-CIT-L-CIT	Vimentin_59–71-Cit_	0.208	0.304	0.226	0.018	39	[29]
4MD4	HLA-DRB1*04:01	1.95	ATEY-CIT-V-CIT-VNSAYQDK	Aggrecan_89–103-Cit_	0.209	0.266	0.197	-0.29	42	[29]
5JLZ	HLA-DRB1*04:01	1.99	TSKGLF(CIR)AAVPSGAS	αEnolasse_26–40-Cit_	0.243	0.389	0.267	0.024	32	[30]
5LAX	HLA-DRB1*04:01	2.60	TSKGLFRAAVPSGAS	αEnolase_26–40_	0.270	0.385	0.298	0.028	20	[30]
5NI9	HLA-DRB1*04:01	1.33	KRIAKAVNEKSCNCL	αEnolase_326–340_	0.176	0.329	0.184	0.008	62	[31]
5NIG	HLA-DRB1*04:01	1.35	K-CIT-IAKAVNEKSCNCL	αEnolase_326–340-Cit_	0.181	0.656	0.178	0.003	92	[31]
6BIJ	HLA-DRB1*04:01	2.10	GGY-CIT-A-CIT-PAKAAAT	Fibrinogen_69–81-Cit_	0.239	0.289	0.220	0.069	36	[32]
6BIL	HLA-DRB1*04:01	2.40	GGYRA-CIT-PAKAAAT	Fibrinogen_69–81-Cit_	0.243	0.305	0.227	0.078	34	[32]
6BIN	HLA-DRB1*04:01	2.50	QYM-CIT-ADQAAGGLR	Collagen_1237–1249-Cit_	0.238	0.319	0.232	0.087	34	[32]
6BIV	HLA-DRB1*04:01	2.90	ETVCP-CIT-TTQQSPE	LL37_86–98-Cit_	0.248	0.469	0.266	0.204	12	[32]
6NIX	HLA-DRB1*04:01	2.10	GIAGFKGEQGPKGEP	Collagen_259–273_	0.235	0.344	0.226	0.118	42	[NA]
4MDI	HLA-DRB1*04:02	2.00	SAVRL-CIT-SSVPGVR	Vimentin_66–78-Cit_	0.203	0.290	0.206	0.003	50	[29]
4MDJ	HLA-DRB1*04:02	1.70	SAVRLRSSVPGVR	Vimentin_66–78_	0.188	0.287	0.190	0.097	70	[29]
4MD5	HLA-DRB1*04:04	1.65	SAVRL-CIT-SSVPGVR	Vimentin_66–78-Cit_	0.186	0.322	0.200	0.014	70	[29]
6BIX	HLA-DRB1*04:04	2.20	ETVCP-CIT-TTQQSPE	LL37_86–98-Cit_	0.213	0.407	0.217	0.190	27	[32]
6BIY	HLA-DRB1*04:04	2.05	DIFERIASEASRL	Histone_70–82_	0.232	0.305	0.224	0.082	56	[32]
6BIZ	HLA-DRB1*04:04	2.10	DIFE-CIT-IASEAS-CIT-L	Histone_70–84-Cit74-Cit81_	0.228	0.322	0.213	0.108	32	[32]
6BIR	HLA-DRB1*04:05	2.30	SSLNL-CIT-ETNLDSL	Vimentin_418–431-Cit423_	0.237	0.322	0.225	0.098	39	[32]
6CPN	HLA-DRB1*11:01	2.00	RFYKTLRAEQASQ	HIV_RQ13_	0.236	0.359	0.235	0.124	42	[14]
6ATZ	HLA-DRB1*14:02	2.70	GGYRA-CIR-PAKAAT	Fibrinogen	0.240	0.334	0.242	0.092	34	[33]
6ATF	HLA-DRB1*14:02	1.90	GVYATRSSAVRLR	Vimentin_59–71_	0.202	0.251	0.201	0.001	60	[33]
6ATI	HLA-DRB1*14:02	1.98	GVYAT-CIR-SSAVRLR	Vimentin_59–71Cit64_	0.233	0.306	0.235	0.002	50	[33]
6CPO	HLA-DRB1*15:02	2.40	RFYKTLRAEQASQ	HIV_RQ13_	0.246	0.332	0.243	0.089	30	[14]
2Q6W	HLA-DRB3*01:01	2.25	AWRSDEALPLGS	Integrin	0.265	0.353	0.249	0.105	34	[34]
3C5J	HLA-DRB3*01:01	1.80	QVIILNHPGQISA	Tu elongation factor	0.227	0.307	0.216	0.091	67	[35]
4H26	HLA-DRB3*03:01	2.50	QWIRVNIPKRI	Synthetic peptide	0.270	0.410	0.264	0.146	24	[36]
1FV1	HLA-DRB5*01:01	1.90	NPVVHFFKNIVTPRTPPPSQ	MBP_119–236_	0.267	0.342	0.252	0.090	43	[37]
1H15	HLA-DRB5*01:01	3.10	GGVYHFVKKHVHES	DNA Pol_628–641_	0.310	0.347	0.280	0.067	12	[38]

**Δ**R_free_ was calculated with Phenix for the R_free_ (refine) and R_free_ (ensemble) values, NA: not available.

**Table 2 ijms-21-07081-t002:** pHLA stability.

pHLA Complex	Tm (°C)
HLA-A2-M1	58.7 ± 0.3
HLA-DR1-CLIP	71.5 ± 0.3
HLA-DR1-RQ13	81.3 ± 0.5
HLA-DR1-HA_306–318_	81.5 ± 0.3
HLA-DR11-CLIP	46.5 ± 0.6
HLA-DR11-RQ13	64.1 ± 1.3
HLA-DR11-HA_306–318_	60.6 ± 0.7

Tm is the temperature required to achieve 50% of unfolded protein, experiment performed at least twice in quadruplicate.

**Table 3 ijms-21-07081-t003:** HLA-DR affinity for the RQ13 peptide.

pHLA Complex	IC_50_ (nM)
HLA-DR11-RQ13	180 ± 50
HLA-DR11-RQ13 + HLA-DM	35 ± 8
HLA-DR1-RQ13	204 ± 30
HLA-DR1-RQ13 + HLA-DM	307 ± 52

The mean and standard error are reported in the table, with the experiment performed at least twice in triplicate.

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
