# Peer review of "Impact of HLA-DR Antigen Binding Cleft Rigidity on T Cell Recognition"

_ijms, 2020, doi:10.3390/ijms21197081_

Round 1
Reviewer 1 Report
This manuscript investigates the dynamics of peptide-HLA class II complex molecules and TCR recognition using MD simulations and ensemble refinement. For this study, the authors utilized previously determined crystal structures of three HLA allomorphs (HLA-DR1, HLA-DR11 and HLA-DR15) presenting the HIV Gag peptide (RQ13) in free state and in complex with a TCR (F24). The study reveals vital observations showing inherent structural rigidity within all three HLA-DR clefts and that HLA-DRβ polymorphisms do not influence the flexibility of cleft. Importantly, the authors identified polymorphic residues that reshape the HLA-DR cleft upon F24 binding. Interestingly, when compared with HLA-DR11, the open cleft conformation of HLA-DR1 did not impact flexibility but associated with higher stability and lower HLA-DM susceptibility. The authors extended the ensemble refinement measurements to available pHLA-DR structures and found that the cleft of majority of HLA-DR molecules are rigid, and implying this phenomenon is not unique to the RQ13 peptide.
The work presented in this manuscript adds significant understanding to the dynamics of the peptide-HLA-DR molecules and its relevance to TCR recognition. The results presented are useful to the wider pHLA/TCR scientific community and I recommended the manuscript for publication with few minor corrections.
- The reader would benefit from moving Figure S1 to the main text
- In page 6 (section 3.3), the sentence “and P5-R forms hydrogen bonds with the carbonyl group of P5-R” should be modified
- Section numbers 3.2-3.5 should be 2.2-2.5 and sections 4 and 5 should be 3 and 4?
- Ensemble refinement was used to show that structural rigidity is a shared feature among all available pHLA-DR complexes. Can authors confirm this also by MD simulations for few selected complexes?
- In Figure S8 legend, please indicate A-C represent different views
- Figure S9(B) legend can be improved for better readability
- The authors touched upon rigidity/flexibility differences between HLA class I and II molecules. This should be elaborated further in the discussion
Author Response
- The reader would benefit from moving Figure S1 to the main text
Although we understand a visual guide for polymorphic residues between HLA allomorphs would be useful as main figure, the Figure S1 is more part of the introduction than result therefore not significant enough to be in the main figure.
- In page 6 (section 3.3), the sentence “and P5-R forms hydrogen bonds with the carbonyl group of P5-R” should be modified
This has been modified to “and Arg71β forms hydrogen bonds with the carbonyl group of P5-R” on page 6 (section 2.3).
- Section numbers 3.2-3.5 should be 2.2-2.5 and sections 4 and 5 should be 3 and 4?
Sections 3.2-3.5 have been renumbered to 2.2-2.5 and sections 4 and 5 and their relevant subheadings have been renumbered to 3 and 4, respectively.
- Ensemble refinement was used to show that structural rigidity is a shared feature among all available pHLA-DR complexes. Can authors confirm this also by MD simulations for few selected complexes?
Consistency between the MD simulations and ER results was demonstrated in the 6 structures, which were studied using both techniques. This supports previous results that show consistency between molecular dynamics and ensemble refinement techniques used in the demonstration of flexibility/rigidity in MHC class I molecules (Fodor et al or reference 13). Given ensemble refinements propensity for demonstrating flexibility and the number of ensemble refinements performed, we believe that there has been ample opportunity for the results to indicate structural flexibility in HLA-DR molecules, if such flexibility was present. The fact that this has not yet been demonstrated then becomes strong evidence that HLA-DR molecules are rigid in the ways that have been described in the manuscript.
In addition, given the current restriction due to COVID-19 pandemic, our lab is currently in lockdown and we would not have the capacity to run MD simulations.
In Figure S8 legend, please indicate A-C represent different views
Figure S8 legend has been changed to include descriptions for Panels A-C: “Each panel shows the same peg-notch interaction from the point of view of (A) the N-terminus, (B) the C-terminus, and (C) a top-down TCR perspective.”
- Figure S9(B) legend can be improved for better readability
Figure S9(B) legend has been changed to make it easier to understand: “(B) The displacement of HATAMRA peptide by increasing concentrations of RQ13 peptide. The shift in binding curves of HLA-DR11 (light blue) indicates that the addition of HLA-DM facilitates displacement of HATAMRA by RQ13 peptide (blue), whereas, HLA-DR1 (light pink) and HLA-DR1 in the presence of HLA-DM (pink) shows no shift.”
- The authors touched upon rigidity/flexibility differences between HLA class I and II molecules. This should be elaborated further in the discussion
Rigidity and flexibility differences between HLA class I and II have indeed been mentioned throughout the manuscript. The majority of these observations on MHC class I conformational plasticity have been derived from our previous work (reference 13, Fodor et al). And as mentioned in our manuscript at Line 122: “We found that rigidity of the cleft was a common trait within all pHLA-DR structures, and in great contrast to the structural plasticity of MHC-I 13.”
In addition, our result contrast with previously observed MHC class I as highlighted in the discussion on page 14 “Stability has been found to be a better predictor of immunogenicity than peptide affinity for MHC class I 12, 19, which in the light of the high stability of HLA-DR1-CLIP may not be the case for HLA-DR1 or for MHC-II molecules in general.”
We have shown that HLA class I and II are different in their rigidity/flexibility, and our study is focused on understanding the link between HLA class II rigidity/flexibility with the different TCR affinity observed in our previous study (reference 14, Galperin et al), and therefore not a comparison between the two classes of HLA molecules.
Reviewer 2 Report
This paper reports very sound and accurate work, in particular because of combining experimental and computational procedures. This review focuses on the computational part of the manuscript.
Regarding molecular dynamics and its evaluation, usual techniques are employed (RMSD, RMSF) to characterize the flexibility of certain atom groups. The authors should add some estimates of RMSF-uncertainties to allow for a judgement if reported differences are substantial (Figure 1, D-H). Although there are only three independent replicates available for each system simulated, some estimate of variability should be provided, e.g. similar to related work by Karch et al in Cells 2019, 8(7), 720, Figures 11 and 12 (https://www.mdpi.com/2073-4409/8/7/720).
The strength of the paper lies in the variety of systems investigated.
Description of MD-simulation methods (lines 238-243) is scarce and should provide more details, e.g. the length of simulation steps, which thermo- and barostats have been employed including their respective time constants, pressure, and temperature. With the amount of information given, it would be hard to reproduce these simulations.
Minor points:
- Typo for unit Ångström in line 239, should read Å.
- Please, provide a definition of the term HLA-DM in the ABSTRACT (last line).
Author Response
Regarding molecular dynamics and its evaluation, usual techniques are employed (RMSD, RMSF) to characterize the flexibility of certain atom groups. The authors should add some estimates of RMSF-uncertainties to allow for a judgement if reported differences are substantial (Figure 1, D-H).
The main outcome for section 2.1 as shown by Figures 1D-H is that the peptide binding cleft for HLA-DR-RQ13 molecules are inherently rigid and that there are no substantial differences in RMSFs across the allomorphs. As the manuscript explores whether the opening of the antigen binding cleft is due to rigidity or flexibility, we only considered differences in rigidity/flexibility relative to each other allomorph, rather than accounting for all regions of flexibility/rigidity within the pMHC molecule. As such, in Figures 1D-F, there are some differences in RMSF distribution between bound and free states of each HLA-DR allomorph, however, when they are grouped into free verses bound simulations (Figure 1G and 1H, respectively), these differences disappear. This is likely due to the inherent flexibility of solvent exposed residues for the free state (Figure 1G), and the constraints placed by the TCR-pMHC interactions in the bound state (Figure 1H).
Therefore, we do not think necessary to show variation for outlier regions as they were either unstructured loop regions, were not located within the antigen binding cleft, or were also consistent with the RMSF magnitudes of the other allomorphs when grouped into free vs bound.
Although there are only three independent replicates available for each system simulated, some estimate of variability should be provided, e.g. similar to related work by Karch et al in Cells 2019, 8(7), 720, Figures 11 and 12 (https://www.mdpi.com/2073-4409/8/7/720).
Figures 11 and 12 from the suggested study by Karch et al., published in Cells 2019, does indeed make fantastic use of visual overlays to quantify RMSF standard deviation. The context of their use is to describe the variability in flexibility of peptide residues and CDR loop residues that underpin TCR-pMHC binding. In our study, we were not able to correlate an open antigen binding cleft with differences in flexibility or rigidity between allomorphs, and therefore, it won’t be helpful to show RMSF variability for specific regions of the antigen binding cleft.
In terms of variability between independent replicates, the degree of variability between simulations can be observed in Figure S2. In the case where an individual residue showed high flexibility, RMSD plots for each independent replicate was also shown (Figure S7). We believe that these plots of independent replicates are showing variability between each simulation relative to significance of the ideas that are being discussed.
On the hand, MD RMSF means were used to describe flexibility for HLA-DRα55, suggesting a site for HLA-DM interaction in line 61-64. These values have been changed to reflect RMSF mean ± SD and have been corrected to:
“The area around HLA-DRα55 reached an RMSF (mean ± SD) of 1.7 ± 0.2 Å in HLA-DR11-RQ13 free, 1.5 ± 0.5 Å in HLA-DR15-RQ13 free, that was decreased to 0.8 ± 0.1 Å and 0.8 ± 0.02 Å, respectively, in both F24 TCR bound complexes (Figure 1G-H). Contrastingly, HLA-DR1 showed a modest reduction in RMSF of 1.0 ± 0.1 Å and 0.7 ± 0.1 Å between free and TCR bound states, respectively (Figure 1G-H).”
The strength of the paper lies in the variety of systems investigated.
Description of MD-simulation methods (lines 238-243) is scarce and should provide more details, e.g. the length of simulation steps, which thermo- and barostats have been employed including their respective time constants, pressure, and temperature. With the amount of information given, it would be hard to reproduce these simulations.
To add detail to the MD protocol used:
The following has been added to line 243-244:
“(300K, 1 atm) via the use of a Berendsen barostat and Langevin thermostat with a damping coefficient of 2 ps-1.”
The following has been added to lines 245-251:
“Energy minimization and equilibration stages consisted of energy minimization via steepest descent followed by conjugate gradient descent until convergence followed by a heat gradient over a 1 ns period with positional restraints on all non-water molecules followed by a 0.2 ns density revision step using isotropic pressure scaling with the same position restraints. The system was then allowed to equilibrate without restraints for 7 ns.
Further information regarding the protocols can be found inside the stageUtil package https://github.com/blake-riley/MD_stageUtil/tree/master/_init/c-Staging/_templates”
Minor points:
- Typo for unit Ångström in line 239, should read Å.
Correction has been implemented in line 239 to read “Å”.
- Please, provide a definition of the term HLA-DM in the ABSTRACT (last line).
HLA-DM is the name of the protein (such as HLA-DR or HLA-DQ), and it is not an acronym.